# Determinants of chronic energy deficiency among non-pregnant and non-lactating women of reproductive age in rural Kebeles of Dera District, North West Ethiopia, 2019: Unmatched case control study

**Asmare Wubie**[1☯]**, Omer Seid**[2☯]**, Sisay Eshetie**[1☯]**, Samuel Dagne**[2]***, Yonatan Menber**[2☯]**, Yosef Wasihun**[3☯]**, Pammla Petrucka**[4‡]

1 Department of Public Health, College of Medicine and Health Sciences, Wollo University, Dessie, Ethiopia, 2 Department of Nutrition and Dietetics, School of Public Health, College of Medicine and Health Science, Bahir Dar University, Bahir Dar, Ethiopia, 3 Department of Health Promotion, School of Public Health, College of Medicine and Health Science, Bahir Dar University, Bahir Dar, Ethiopia, 4 University of Saskatchewan, Saskatoon, SK, Canada

☯ These authors contributed equally to this work.
‡ PP also contributed equally to this work.
* samueldagne1888@gmail.com

## Abstract

### Background

In Ethiopia about 25% of rural women are chronically malnourished. Non-pregnant and non-lactating women present an opportunity to implement strategies to correct maternal and child health status and to potentiate improved pregnancy outcomes in developing countries like Ethiopia. The determinant factors of chronic energy deficiency vary across settings and contexts; hence, it is important to identify local determinant factors in order to implement effective and efficient intervention strategies.

### Objective

To assess the determinants of chronic energy deficiency of non-pregnant, non-lactating rural women within the reproductive age group (15–49 years), in rural kebeles of Dera district, North West Ethiopia, 2019.

### Methods

A community based unmatched case control study was conducted. A total of 552 participants were involved and a multi-stage sampling technique was used to select the samples. Data was collected from January 15 to February 30, 2019 using face-to-face interviews and anthropometric assessments. EPI-info version 7 and SPSS™ version 23 were used for data entry and analysis, respectively. Bivariable and multivariable logistic regression models were used to analyze the association between dependent and independent variables. Association was considered statistically significant at 95% CI with p-value < 0.05 in multivariable logistic regression.

**Data Availability Statement:** All relevant data are within the paper and its Supporting Information files.

**Funding:** The author(s) received no specific funding for this work.

**Competing interests:** The authors have declared that no competing interests exist.

**Abbreviations:** ANC, Ante Natal Care; BMI, Body Mass Index; CED, Chronic Energy Deficiency; DDs, Dietary Diversity Score; EDHS, Ethiopian Demographic Health Survey; ETB, Ethiopian Birr; FAO, Food and Agriculture Organization; NNP, National Nutrition Program; NPNL, Non Pregnant Non Lactating; SPSS, Statistical Packages for Social Sciences; WHO, World Health organization.

## Result

A total of 548 non-pregnant, non-lactating women with 137 cases and 411 controls were included in the study with a response rate of 99.3%. High family size (AOR = 1.88, 95% CI: 1.085, 3.275), low educational status (AOR = 3.389, 95% CI: 1.075, 10.683), inadequate meal frequency (AOR = 5.345, 95% CI: 2.266, 12.608), absence of home garden (AOR = 5.612, 95% CI: 3.177, 9.915) and absence of latrine facility (AOR = 6.365, 95% CI: 3.534, 11.462) were found positively associated with chronic energy deficiency.

## Conclusion and recommendation

Inadequate meal frequency, absence of home gardening, absence of latrine facility, high family size and educational status of illiterate were the determinants of chronic energy deficiency, thus indicating the imperative for a multi-sectoral approach with health, agriculture and education entities developing and delivering interventions.

## Background

Chronic energy deficiency (CED) is a condition of a body characterized by low body weight and low energy stores, possibly limited physical capacity due to deprivation of food over a long period of time with body mass index (BMI) less than 18.5 kg/m$^2$ for adults [1–3]. Chronic energy deficiency is higher among rural women of reproductive age and caused by eating too little or having an unbalanced diet that lacks adequate nutrients.

CED before pregnancy causes major perinatal risks like stillbirths, preterm births, and small for gestational age and low birth weight babies [4–7]. It leads to low productivity among adults and is related to heightened morbidity and mortality [6, 8, 9]. Children of undernourished women are more likely to be undernourished, which can lead to poor cognitive development, shorter stature, and higher risk of morbidity and mortality [9–12].

Under nutrition among non-pregnant women and non-lactating (NPNL) mother is one of the serious public health problems in developing world [1, 13, 14]. Maternal and child under nutrition is the underlying cause of at least 3.5 million deaths each year and constitutes 11% of the total global disease burden [15].

The United Nations Food and Agriculture Organization (FAO) estimated that nearly one in eight people were suffering from chronic undernourishment in 2010–2012 with most occurring in developing countries. About 75% of the world's poor reside in rural areas [16], with Ethiopia, as one of the most populous countries in Africa, having 84% rural population [17].

CED among NPNL women is a major public health problem in Ethiopia, particularly higher in rural parts. The problem is more prevalent in Ethiopia compared with other African countries like Botswana and Tanzania [18, 19].

The past consecutive Ethiopian Demographic Health Surveys (EDHS) showed that a marginal decline in the magnitude of CED among NPNL women (aged 15–49 years) at 30.5%, 26.9%, 27% and 21.4% in 2000, 2005, 2011, and 2016 EDHS, respectively [13].

To address such problems Ethiopia has adopted the national nutrition program (NNP) to empower women, and development partners are committed to improving the nutritional impacts of women in the country [20].

Different studies showed that the determinant factors of CED varied [18, 21]. Factors, such as age, occupation, meal frequency, health status, marital status, educational status, residence,

number of parity and food insecurity, are some of the causes of CED in non-pregnant women [18, 21–23].

This study focused on the Amhara Region of Ethiopia, which according to EDHS 2016, had a reported prevalence of CED of 23% [24]. Among the regions, women in Amhara, were the fifth most affected by chronic energy deficiency [13, 20, 24–27]. So the aim of conducting this study in Dera District was to determine specific local determinants of CED among NPNL women of a rural area. Knowing these determinant factors may inform effective and efficient intervention strategies. So conducting this research was important to plan and develop prevention strategies of chronic energy deficiency among NPNL women that are specific and responsive to women in the Amhara Region.

## Methods and materials

### Study area and period

This study was conducted in rural kebeles of Dera District, South Gondar Zone between January15 to February 28/2019. Dera District is one of the 15 districts in South Gondar Zone, Amhara Regional State, Ethiopia. Dera Districts is located at 608 km northwest of Addis Ababa, the capital city of Ethiopia and 42 km from Bahir Dar capital city of Amhara region. The District has a total population of 293,071 of a 2011 Ethiopian physical year projection (male 147,122 and female 145,949) in 68,140 households. Among the total population 67,832 were non-pregnant women in the reproductive age group. It is subdivided into 11 clusters /37 Kebeles/ with a specified number of Kebeles /local administrative units/ and health posts for each cluster. There are 11 public health centers and 38 health posts with at least 2 health extension workers in each community health post to provide basic primary health care services [28]. Agricultural products like maize, millet, wheat, teff, sorghum and fruits like mango, banana and orange are cultivated in most parts of the District.

### Study design

A community based unmatched case control study design was employed.

### Inclusion and exclusion criteria

All NPNL women in the reproductive age group (15–49 years) who have lived within the Amhara Region for more than 6 months were included in the sample. Women with mental problems and/ or were unable to communicate during data collection were excluded from the study.

### Sample size determination

The sample size was calculated using Epi Info version 7.2.1.1 by considering the following assumptions: The proportion of individuals' who spent greater than 30 minute to fetch drink water was 38.2% among controls exposed and 59.4% among cases exposed [29], 95% Confidence level, 80% power, case to control ratio of 1:3, 10% for non-response rate & design effect of 2. The total sample size was 552 (138 cases and 414 controls).

### Sampling procedure

A multistage sampling technique was used to conduct this study. In the first stage, from a total of 37 kebeles in the district, seven kebeles were selected using lottery method. Then, the total sample size was allocated proportionally to each kebele. Preliminary survey was conducted and anthropometric data was obtained from 6506 NPNL women in the selected Kebles. After cases and controls were identified, NPNL women had got identification number registered

sequentially. Finally, both cases and controls were selected by simple random sampling from the preliminary censuses data by computer generating system.

## Data collection tools and procedures

The data was collected using interview administered structured and pre-tested questionnaire which includes socio-demographic, hygiene and sanitation, maternal and diet related factors adapted from previous studies [13, 18, 21, 29] by well trained and experienced ten diploma nurse data collectors and three BSc nurse supervisors.

The Dietary Diversity Score (DDS) was calculated from a single 24 hour recall prior to data collection. All foods consumed in a day before study was grouped in- to nine categories and consuming a food item from any of the groups was assigned a score of 1 and if no food was taken a score of 0 was given. Accordingly, a DDS of up to 9 points was computed by adding the scores which were classified as low ($\leq 3$), medium (4–6) and high (7–9) [30].

Nutritional status of non- lactating and non-pregnant women was screened by measuring height and Weight and calculating BMI. Height of women aged 15–49 years and who are NPNL was measured using height scale. The women standing upright with bare foot and the women's heads, shoulders, buttocks, knees and heels were made to touch the height scale. The reading was recorded to the nearest 0.1cm. Weight of study participants were measured with minimum/light/clothing and no shoes with the reading recorded to the nearest 0.1 kg [31]. Women with a BMI of less than 18.5kg/m2 were considered as exhibiting chronic energy deficiency [1, 32].

## Data quality control

The questionnaire was translated to local language (Amharic) and back to English for consistency (See S1 File). Pre-testing was done within 28 individuals at a place where the actual data collection was not conducted. Data collectors and supervisors were trained for 2 days. Calibration of instruments was conducted before every women measurement of weight by weighing standard weight and by using standard measuring instrument. On spot checking and correction was made for incomplete questionnaires by the supervisor. The overall data collection process was controlled by the principal investigator.

## Data analysis

The data was coded, cleaned and entered into Epi Info Version 7 and exported to SPSS™ version 23 for analysis. Outcome variable was dichotomized into 1 = cases and 0 = controls. Descriptive statistics were computed and the result was reported using tables, figures and charts. Bi-variable logistic regression was executed and variables with $p < 0.25$ were fitted to the final multivariable logistic regression to adjust for potential confounders to identify the determinants of chronic energy deficiency among none pregnant and none lactating women. In the final model, variables with P-value $< 0.05$ were considered as statistically significant and AOR of 95% CI was used to see the strength of association. Multicollinearity between the independent variables was also assessed using multiple linear regressions. No evidence of multicollinearity was found as the variance inflation factor (VIF) for all variables was less than 10. Model fitness was checked by Hosmer& Lemshow test and it was non-significant with p-value of 0.4, which show that the model was fit.

## Ethics approval and consent to participate

Ethical clearance was obtained from the institutional review board of Wollo University. A permission letter was obtained from Dera District Health Office and from kebele leaders. Further,

study participants were briefed about the main objective of the study. Participants were informed that they have the full right to refuse to participate in the study or can interrupt/withdraw if they want. Confidentiality of the information was assured and the privacy of the study participants was respected and kept as well. Written informed consent was obtained from each study participant and/or from parents/guardians of <18 years old study participants. At the last nutritional counseling were given to women who were chronic energy deficient and over weighted.

## Results

### Socio-demographic characteristics of respondents

A total of 548 NPNL women with 137 cases and 411 controls were included in the study with a response rate of 99.3%. The mean age of the study participants was 29.4 ± SD 6.73 years of age. About 485(88.5%) of the respondents were married. About two thirds, 354 (64.6%) of respondents cannot read and write (Table 1).

### Hygiene and sanitation characteristics of characteristics

Among the respondents, 274(50%) used open defecation. Only 23 (16%) of the cases and 251 (61%) of the controls used latrine. Only 184(33.6%) of the respondents used water from a safe

**Table 1. Socio demographic and economic characteristics of non-pregnant non lactating women of reproductive age in rural Dera District population, Northwest Ethiopia, 2019.**

| Variables | Cases | Controls | Total N (%) |
|---|---|---|---|
|  | Frequency (%) | Frequency (%) |  |
| **Age of respondents** |  |  |  |
| 15–24 | 25(18.2) | 102(24.9) | 127(23.2) |
| 25–34 | 63(46.0) | 199(48.4) | 262(47.8) |
| 35–49 | 49(35.8) | 110(26.7) | 159(29.0) |
| **Marital status** |  |  |  |
| Married (living with partner) | 123(89.8) | 362(88.1) | 485(88.5) |
| Living with no partner* | 14(10.2) | 49(11.9) | 63(11.5) |
| **Educational status of women** |  |  |  |
| Cannot write and read | 99(72.3) | 258(62.8) | 357(65.2) |
| Can write and read | 32(23.4) | 70(17.0) | 102(18.6) |
| Primary education above | 6(4.4) | 83(20.2) | 89(16.2 |
| **Head of house hold** |  |  |  |
| Husband | 126(92.0) | 376(91.5) | 502(91.6) |
| Women | 11(8.0) | 35(8.5) | 46(8.4) |
| **Educational status of house hold head** |  |  |  |
| Cannot write and read | 103(75.2) | 251(61.1) | 354(64.6) |
| Can write and read | 29(21.2) | 121(29.4) | 150(27.4) |
| Primary education and above | 5(3.6) | 39(9.5) | 44(8.0) |
| **Monthly income of the house hold** |  |  |  |
| <1000 ETB | 86(62.8) | 170(41.4) | 256(46.7) |
| 1000–2500 ETB | 46(35.6) | 218(53.0) | 264(48.2) |
| >2500 ETB | 5(3.6) | 23(5.6) | 28(5.1) |
| **Family size** |  |  |  |
| <5 | 69(50.4) | 284(69.1) | 353(64.4) |
| ≥5 | 68(49.6) | 127(30.9) | 195(36.6) |

source (protected well and tap water) with a mean fetch time to collect water in minute of 14.56 ± 10.8 (SD) (Table 2).

### Women reproductive history and respondent characteristics on illness

Among 548 women, about 319 (60.4%) respondents were married before the age of 18 years and 20 of the respondents were single in marital status. About 449 (81.9%) used family planning and the coverage were 81.8% among cases and 82% from controls (Table 3).

### Dietary habit of respondents

About 503(91.8%) of respondents reported eating three or more meals per day regularly which was 78.8% among cases and 96.1% among controls. About 207 (58.5%) of respondents had no home gardening (Table 4).

### Diet diversity score of respondents

Two thirds of the respondents 346(63.1%) had medium DDS from the 9 food groups in the past 24 hours, while 178(32.5%) of the women had low DDS (Fig 1).

## Determinants of CED among non-pregnant women & non-lactating mother

The result of bi-variable analysis showed a significant association between CED and age at first marriage, family size, women diet diversity, presence of home garden, women meal frequency, latrine facility, history of ANC follow up, educational status of the women and the house hold head at a p-value of 0.25.

**Table 2. Hygiene and sanitation characteristics of NPNL women in the reproductive age group in Dera population, Northwest, Ethiopia, February 2019.**

| Variables | cases | Controls | Total N (%) |
|---|---|---|---|
|  | Frequency (%) | Frequency (%) |  |
| **Latrine facility** |  |  |  |
| Yes | 23(16.8) | 251(61.1) | 274(50.0) |
| No | 114(83.2) | 160(38.9) | 274(50.0) |
| **Latrine utilization** |  |  |  |
| Yes | 23(16.8) | 249(60.6) | 272(49.6) |
| No | 114(83.2) | 162(39.4) | 276(50.40 |
| **Hand washing after latrine** |  |  |  |
| Yes | 20(14.6) | 163(39.7) | 183(33.4) |
| No | 117(85.4) | 248(60.3) | 365(66.6) |
| **Drinking water source** |  |  |  |
| Protected | 41(29.9) | 143(34.8) | 184(33.5) |
| Un protected | 96(70.1) | 268(65.2) | 364(66.4) |
| **Time to fetch drink water** |  |  |  |
| ≤30minute | 131(95.6) | 389(94.6) | 520(94.9) |
| >30 minute | 6(4.4) | 22(5.4) | 28(5.1) |
| **Treatment of water** |  |  |  |
| Yes | 5(3.6) | 40(9.0) | 45(8.2) |
| No | 132(96.4) | 371(91.0) | 503(91.8) |

**Table 3. Illness and reproductive history of NPNL women in the reproductive age group in rural Dera population, Northwest, Ethiopia, February 2019.**

| Variables | Cases | Controls | Total N (%) |
|---|---|---|---|
|  | Frequency (%) | Frequency (%) |  |
| **Age at first marriage** |  |  |  |
| ≥ 18yrs | 45(33.6) | 164(41.6) | 209(39.6) |
| <18yrs | 89(66.4) | 230(58.4) | 319(60.4) |
| **Family planning use** |  |  |  |
| Yes | 112(81.8) | 337(82.0) | 449(81.9) |
| No | 25(18.2) | 74(18.0) | 99(18.1) |
| **Gravidity** |  |  |  |
| 1–2 | 38(30.6) | 149(41.0) | 187(38.4) |
| 3–4 | 41(33.1) | 105(28.9) | 146(30.0) |
| ≥5 | 45(36.3) | 109(30.0) | 154(31.6) |
| **Parity** |  |  |  |
| 1–2 | 41(33.1) | 155(43.1) | 196(40.5) |
| 3–4 | 44(35.4) | 108(30.0) | 152(31.4) |
| ≥5 | 39(31.5) | 97(26.9) | 136(28.1) |
| **History of ANC follow up** |  |  |  |
| Yes | 101(73.7) | 326(89.8) | 427(87.7) |
| No | 23(16.8) | 37(10.2) | 60(12.3) |
| **History of illness in the past 1month** |  |  |  |
| Yes | 5(3.6) | 15(3.6) | 20(3.6) |
| No | 132(96.4) | 396(96.4) | 528(96.4) |

However, in multivariable logistic regression analysis only family size, absence of home garden, low educational status of women, absence of latrine facility, and inadequate meal frequency was significantly associated with CED.

NPNL women, who cannot read and write, were 3.4 times more likely to be chronic energy deficiency than women who completed primary education (AOR = 3.39; 95%CI: 1.08, 10.68). NPNL women who had no home garden were 5.6 times more likely to be chronically energy deficient than women who had home gardens (AOR = 5.61; 95%CI: 3.18, 9.92). NPNL women with meal frequency <3 meals per day were more than 5 times more likely to be chronic energy deficient than women with meal frequency ≥3 meals per day (AOR = 5.35; 95%CI: 2.27, 12.61). NPNL women without latrine facilities had 6.4 times higher odds of CED than women who had latrine facility (AOR = 6.37; 95%CI: 3.53, 11.46). According to this study, NPNL women with family size ≥5 were 1.89 times more likely to be chronically energy deficient than those with family size less than five (AOR = 1.89; 95% CI: 1.09, 3.28) (Table 5).

**Table 4. Dietary habits of NPNL women in the reproductive age group in Rural Dera population, Northwest, Ethiopia, February 2019.**

| Variables | Cases | Controls | Total N % |
|---|---|---|---|
|  | Frequency (%) | Frequency (%) |  |
| **Meal frequency per day** |  |  |  |
| ≥ 3 meals per day | 108(78.8) | 395(96.1) | 503(91.8) |
| < 3 meals per day | 29(21.2) | 16(3.9) | 45(8.2) |
| **Presence of home gardening** |  |  |  |
| Yes | 30(21.9) | 251(61.1) | 281(51.3) |
| No | 107(78.1) | 160(38.9) | 267(48.7) |

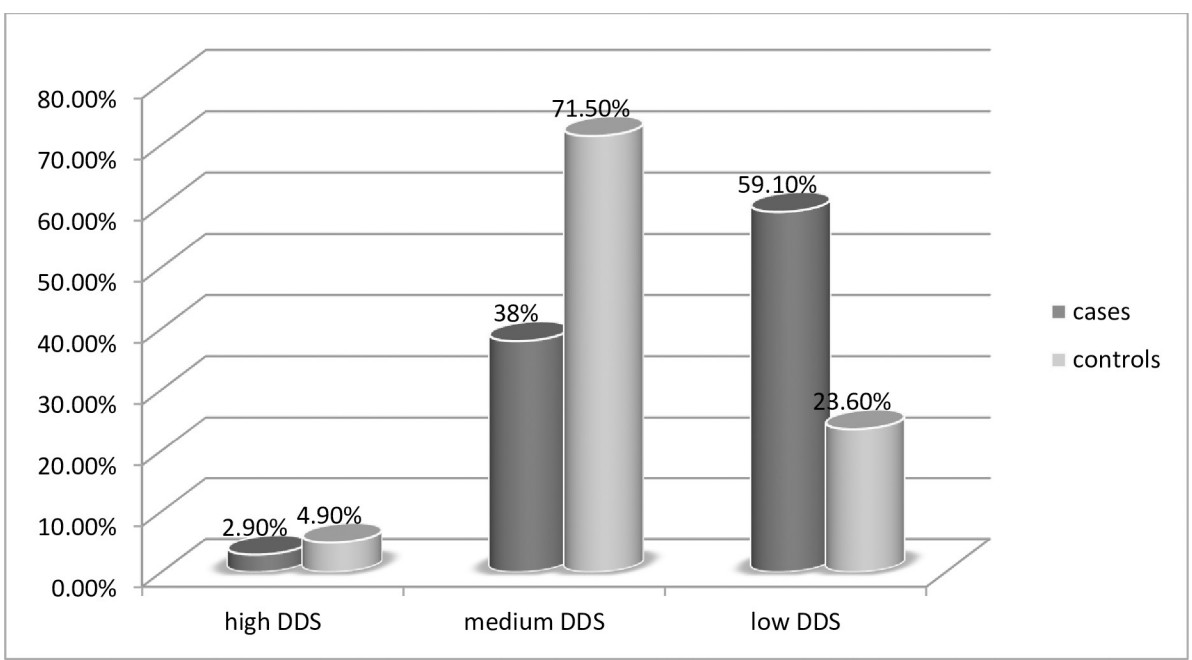

**Fig 1. Diet diversity score of non-pregnant non-lactating women in rural kebeles of Dera District, February 2019.**

## Discussion

Under nutrition is a critical condition, and it is evident that the condition is higher among rural women of reproductive age, which is the single largest contributor to the burden of disease in low and middle income countries impacting social and economic development. CED among women is also a major risk factor for adverse birth outcomes. Although reducing under nutrition with a purview to addressing the determinants of chronic energy deficiency, progress in reducing under nutrition remains slow in Ethiopia [14, 29, 33]. According to this study, statistically significant predictor variables were high family size, educational status of women, meal frequency, absence of home gardens, and absence of latrine facilities.

NPNL women with meal frequency <3 meals per day were more than 5 times affected by chronic energy deficiency than that of women with meal frequency≥3 meals per day. This positively associated result was consistent with studies conducted in Kunama (Tigray), Debretabor (Amhara) & Asayita (Afar) respectively [21, 22, 29]. However, meal frequency did not show statistical significant association with under nutrition in a study done among Bangladesh women of reproductive age [34]. This variation may be due the residence and feeding habit differences.

Another important determinant of CED was women's educational status. NPNL women who cannot read &write, were 3.4 times more likely to be chronic energy deficient than women who complete primary education and above. This trend may reflect how women without formal education may not have awareness about health & nutrition related issues which undermine their wellbeing and nutritional status. Worldwide studies also showed that being educated is negatively associated with underweight and positively associated to be overweight [18, 35–38].

In India and Bangladesh rural women without formal education were more affected by chronic energy deficiency than women with primary education and above [14, 38]. Similarly studies in African countries like Tanzania, Botswana and Arssi, Ethiopia [18, 19, 23]. However,

**Table 5. Logistic regression of nutritional status (BMI) and predictors among women of reproductive age in rural Dera District population, Northwest, Ethiopia, 2019.**

| Explanatory Variables | Nutritional status | | Odds ratio (95% CI) | |
| --- | --- | --- | --- | --- |
| | Cases | Control | COR | AOR |
| | | | | |
| **Age at first marriage** | | | | |
| ≥18 | 45(33.6%) | 164(41.6%) | 1 | 1 |
| <18 | 89(66.4%) | 230(58.4%) | 1.41(1.14, 2.13) | 1.14(0.65, 1.99) |
| **Family size** | | | | |
| <5 family | 69(50.4%) | 284(69.1%) | 1 | 1 |
| ≥5 family | 68(49.6%) | 127(30.9%) | 2.20(1.49, 3.27) | 1.89(1.09, 3.28) |
| **DDS** | | | | |
| Low | 79(57.6%) | 97(23.6%) | 4.18(1.37, 12.71) | 3.25(0.79, 13.42) |
| Medium | 52(38.0%) | 294(71.6%) | 0.88(0.29, 2.69) | 0.67(0.16, 2.74) |
| High | 6(4.4%) | 20(4.8%) | 1 | 1 |
| **Home garden** | | | | |
| Yes | 30(21.9%) | 251(61.1%) | 1 | 1 |
| No | 107(78.1%) | 160(38.9%) | 5.59(3.57, 8.78) | 5.61(3.18, 9.92) |
| **Number of meal per day** | | | | |
| ≥3 meals | 108(78.8%) | 395(96.1%) | 1 | 1 |
| <3 meals | 29(21.2%) | 16(3.9%) | 6.63(3.47, 12.65) | 5.35(2.27, 12.61) |
| **Latrine facility** | | | | |
| Yes | 23(16.8%) | 251(61.1%) | 1 | 1 |
| No | 114(83.2%) | 160(38.9%) | 7.78(4.76, 12.69) | 6.37(3.53, 11.46) |
| **Educational status** | | | | |
| Cannot write read | 99(77.3%) | 258(62.8%) | 5.31(2.25, 12.55) | 3.39(1.08, 10.68) |
| Can write & read | 32(23.4%) | 70(17.0%) | 6.32(2.50, 15.99) | 4.12(1.19, 14.16) |
| Primary & above | 6(4.3%) | 83(20.2%) | 1 | 1 |
| **History of ANC** | | | | |
| Yes | 101(81.4%) | 326(89.3%) | 1 | 1 |
| No | 23(18.6%) | 39(10.7%) | 1.90(1.09, 3.34)* | 0.86(0.40, 1.85) |

studies conducted in Tigray Region, Kunama Districts showed that education was not statistically -associated with underweight [29], which might reflect variance in population which included both urban and rural women.

NPNL women without access to latrine facilities had 6.4 times higher odds of CED than women who had latrine facility access. This finding may be due to women lacking latrines being exposed to fecal-oral infections which, in turn, lead to under nutrition. Similar findings were reported from studies conducted in Addis Ababa and Arssi, Ethiopia [20, 23].

Another statistically significant environmental determinant was availability of home gardens. NPNL women who had no home gardens were 5.6 times more likely to be CED than women who had home gardens. This might be due to women who had home gardening would have chance of eating diversified food regularly and may contribute to appropriate weight. This finding was consistent with prior studies in Cambodia, Nepal, and Philippines [39, 40].

According to this study, Non pregnant and non-lactating women with family size ≥5 were 1.88 times more likely to be chronically energy deficient than those with family size less than five. High family size in rural communities may increase women's work load. In Ethiopian, rural women have role as mothers and housewives, with a range of responsibilities like food preparation, agricultural activities, fetching water, and wood for cooking. This intense

workload might increase the energy consumption of the body leading to exhaustion. This study is consistent with previous research conducted in China and Oromia Region of Ethiopia [41, 42]. The present study has some limitation. First, portion size of the meal was not assessed which may have added a measure of the actual amount of food consumed by NPNL women take. Second, this study used WHO BMI cut off points which may lead it to misclassification.

## Conclusion

High family size, inadequate women meal frequency, low women educational status, absence of home garden & absence of latrine facility were determinant factors that increase chronic energy deficiency of NPNL women in rural Kebeles of Dera District. Thus multi- sectorial approach intervention with agriculture and education offices should be considered to tackle CED. We also recommend other researchers to consider sustained research agenda which included other researchers, better to address socio- cultural issues related to nutrition.

## Supporting information

**S1 File.**
(DOCX)

## Acknowledgments

Authors would like to thanks Wollo University for approving ethical clearance. We would like also to thank data collectors, supervisors and study participants.

## Author Contributions

**Conceptualization:** Asmare Wubie, Omer Seid, Sisay Eshetie, Samuel Dagne, Yonatan Menber, Yosef Wasihun, Pammla Petrucka.

**Data curation:** Omer Seid, Pammla Petrucka.

**Formal analysis:** Asmare Wubie, Omer Seid, Yonatan Menber.

**Investigation:** Asmare Wubie.

**Methodology:** Asmare Wubie, Omer Seid, Sisay Eshetie.

**Project administration:** Omer Seid, Samuel Dagne, Yonatan Menber, Yosef Wasihun, Pammla Petrucka.

**Software:** Asmare Wubie, Omer Seid, Sisay Eshetie, Samuel Dagne.

**Supervision:** Asmare Wubie.

**Validation:** Omer Seid, Sisay Eshetie, Samuel Dagne, Yonatan Menber, Yosef Wasihun, Pammla Petrucka.

**Writing – original draft:** Asmare Wubie, Samuel Dagne.

**Writing – review & editing:** Omer Seid, Sisay Eshetie, Samuel Dagne, Yonatan Menber, Yosef Wasihun, Pammla Petrucka.

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
