## [Decision Letter · Decision Letter 0]

17 Sep 2020

PONE-D-20-26032

Determinants of chronic energy deficiency among non-pregnant and non-lactating women of reproductive age in rural Kebeles of Dera district, North West Ethiopia, 2019: Unmatched case control study

PLOS ONE

Dear Dr. Dagne,

Thank you for submitting your manuscript to PLOS ONE. After careful consideration, we feel that it has merit but does not fully meet PLOS ONE’s publication criteria as it currently stands. Therefore, we invite you to submit a revised version of the manuscript that addresses the points raised during the review process.

An expert in the field handled your manuscript, and we are very thankful for their time and efforts. Although interest was found in your study, some comments arose that require your attention. Please address ALL of the reviewer's comments in your revised manuscript.

We look forward to receiving your revised manuscript.

Kind regards,

Frank T. Spradley

Academic Editor

PLOS ONE

3. In the Methods, please discuss whether and how the questionnaire was validated. If this did not occur, please provide the rationale for not doing so.

4. Please discuss the following publication in your manuscript: https://bmcnutr.biomedcentral.com/articles/10.1186/s40795-015-0005-y

5. Please provide additional details regarding participant consent. In the ethics statement in the Methods and online submission information, please ensure that you have specified what type you obtained (for instance, written or verbal, and if verbal, how it was documented and witnessed). If your study included minors, state whether you obtained consent from parents or guardians. If the need for consent was waived by the ethics committee, please include this information.

6. Please amend either the abstract on the online submission form (via Edit Submission) or the abstract in the manuscript so that they are identical.

7. Your ethics statement should only appear in the Methods section of your manuscript. If your ethics statement is written in any section besides the Methods, please move it to the Methods section and delete it from any other section. Please ensure that your ethics statement is included in your manuscript, as the ethics statement entered into the online submission form will not be published alongside your manuscript.

Reviewers' comments:

Reviewer's Responses to Questions

**Comments to the Author**

1. Is the manuscript technically sound, and do the data support the conclusions?

Reviewer #1: Yes

2. Has the statistical analysis been performed appropriately and rigorously? 

Reviewer #1: I Don't Know

3. Have the authors made all data underlying the findings in their manuscript fully available?

Reviewer #1: Yes

4. Is the manuscript presented in an intelligible fashion and written in standard English?

Reviewer #1: Yes

5. Review Comments to the Author

Reviewer #1: Line 120-121: what is the basis for considering women with a BMI of less than 18.5 km/m2 as chronic energy deficiency. Are there any references?

Table 4, Does presence of home gardening affect dietary habits?

Did the author collect other dietary habits such as hand washing before eating, with regular quantitative meal daily or not?

6. PLOS authors have the option to publish the peer review history of their article (what does this mean?). If published, this will include your full peer review and any attached files.

Reviewer #1: No

---

## [Author Response · Author response to Decision Letter 0]

5 Oct 2020

Response to Reviewers

Academic editor:

1. Please discuss whether and how the questionnaire was validated. If this did not occur, please provide the rationale for not doing so.

- The questionnaire was adapted after reviewing previous literatures done on similar title. We have cited the references in line -109. Anthropometric measurements (height and weight) were done based on WHO standards (line 121). DDS of the study participants was collected based on Food and agriculture organization (FAO) guide line to measurement minimum dietary diversity for women(line 111- 115).

2. Discuss the following publication: https://bmcnutr.biomedcentral.com/articles/10.1186/s40795-015-0005-y.

The above publication is used and discussed in the manuscript. Check line 226. In this study, meal frequency was one determinant factor for CED among study participants.

3. Regarding participant consent (<18 years), Participants were informed that they have the full right to refuse to participate in the study or can interrupt/withdraw if they want. Confidentiality of the information was assured and the privacy of the study participants was respected and kept as well. Written informed consent was obtained from each study participant and/or from parents/guardians of <18 years old study participants (line 148-152).

Reviewer #1:

1. Line 120-121: what is the basis for considering women with a BMI of less than 18.5 kg/m2 as chronic energy deficiency? Are there any references?

- According to the report of a working party of the International Dietary Energy Consultative Group, BMI of less than 18.5 kg/m2 among adults is considered as chronic energy deficiency. I have cited the reference in line 122-123.

2. Tables 4, Does presence of home gardening affect dietary habits? Did the author collect other dietary habits such as hand washing before eating, with regular quantitative meal daily or not?

- We have check multicollinearity between all independent variables. Variance inflation factor was (VIF) below 10. Therefore, presences of home gardening do not affect dietary habits (line 140-142). 

- We did not collect information on hand washing before eating and portion size of meal. We have mentioned these as a limitation (line 258-260). We rather collect information on access to safe drinking water and latrine facility including hand washing after toilet. We indirectly try to assess the hand washing habit of the study participants (S3- questionnaire).

---

## [Decision Letter · Decision Letter 1]

14 Oct 2020

Determinants of chronic energy deficiency among non-pregnant and non-lactating women of reproductive age in rural Kebeles of Dera district, North West Ethiopia, 2019: Unmatched case control study

PONE-D-20-26032R1

Dear Dr. Dagne,

We’re pleased to inform you that your manuscript has been judged scientifically suitable for publication and will be formally accepted for publication once it meets all outstanding technical requirements.

Kind regards,

Frank T. Spradley

Academic Editor

PLOS ONE

Reviewers' comments:

Reviewer's Responses to Questions

**Comments to the Author**

1. If the authors have adequately addressed your comments raised in a previous round of review and you feel that this manuscript is now acceptable for publication, you may indicate that here to bypass the “Comments to the Author” section, enter your conflict of interest statement in the “Confidential to Editor” section, and submit your "Accept" recommendation.

Reviewer #1: All comments have been addressed

2. Is the manuscript technically sound, and do the data support the conclusions?

Reviewer #1: Yes

3. Has the statistical analysis been performed appropriately and rigorously? 

Reviewer #1: I Don't Know

4. Have the authors made all data underlying the findings in their manuscript fully available?

Reviewer #1: Yes

5. Is the manuscript presented in an intelligible fashion and written in standard English?

Reviewer #1: Yes

6. Review Comments to the Author

Reviewer #1: the authors have adequately addressed your comments raised in a previous round of review and I have no further comments.

7. PLOS authors have the option to publish the peer review history of their article (what does this mean?). If published, this will include your full peer review and any attached files.

Reviewer #1: No

---

## [Editor Report · Acceptance letter]

19 Oct 2020

PONE-D-20-26032R1 

Determinants of chronic energy deficiency among non-pregnant and non-lactating women of reproductive age in rural Kebeles of Dera district, North West Ethiopia, 2019:  Unmatched case control study 

Dear Dr. Dagne:

I'm pleased to inform you that your manuscript has been deemed suitable for publication in PLOS ONE. Congratulations! Your manuscript is now with our production department. 

Kind regards, 

on behalf of

Dr. Frank T. Spradley 

Academic Editor

PLOS ONE